# Bronchial Carcinoids: From Molecular Background to Treatment Approach

**DOI:** 10.3390/cancers14030520

**Published:** 2022-01-20

**Authors:** Marta Araujo-Castro, Eider Pascual-Corrales, Javier Molina-Cerrillo, Nicolás Moreno Mata, Teresa Alonso-Gordoa

**Affiliations:** 1Neuroendocrinology Unit, Endocrinology and Nutrition Department, Hospital Universitario Ramón y Cajal, 28034 Madrid, Spain; eider.pascual@salud.madrid.org; 2Instituto de Investigación Biomédica Ramón y Cajal (IRICYS), 28034 Madrid, Spain; talonso@salud.madrid.org; 3Universidad de Alcalá, 28801 Madrid, Spain; 4Medical Oncology Department, Hospital Universitario Ramón y Cajal, 28034 Madrid, Spain; 5Thoracic Surgery Department, Hospital Universitario Ramón y Cajal, 28034 Madrid, Spain; nicolas.moreno.hrc@gmail.com

**Keywords:** bronchial carcinoid, lung neuroendocrine tumors, multiple endocrine neoplasia type 1, everolimus, somatostatin analogues

## Abstract

**Simple Summary:**

Bronchial carcinoids (BCs) are uncommon and usually slow growing neuroendocrine epithelial malignancies that represent less than 2% of all lung cancers. Differences in the extent of molecular alterations between neuroendocrine carcinomas and BCs may underline the differences in the aggressiveness of these lesions. Moreover, although atypical BCs and typical BCs have similar set of mutations, some differential molecular and genetic alterations have been described between these two entities. A better understanding of the genetic and molecular background of BCs would allow a better selection of medical treatments in these patients. Regarding treatment, most BCs can be cured by surgery; however, inoperable tumors are mostly insensitive to chemotherapy and radiotherapy. In advanced BCs, the only drug that has a positive phase III clinical trial in BCs is everolimus. Somatostatin analogues constitute the gold standard for symptomatic relief. Peptide receptor radionuclide therapy has been associated with longer progression free. The efficacy of other treatments such as antiangiogenic agents and immunotherapy is still not established.

**Abstract:**

A better understanding of the genetic and molecular background of bronchial carcinoids (BCs) would allow a better estimation of the risk of disease progression and the personalization of treatment in cases of advanced disease. Molecular studies confirmed that lungs neuroendocrine tumors (NETs) and neuroendocrine carcinomas (NECs) are different entities; thus, no progression of NET to NEC is expected. In BCs, MEN1 gene mutations and deletions and decreased gene expression have been associated with a poor prognosis. ATRX mutation has also been linked to a shorter disease-specific survival. In terms of therapeutic targets, PI3K/AKT/mTOR pathway mutations have been described in 13% of typical carcinoids (TCs) and 39% of atypical carcinoids (ACs), representing a targetable mutation with kinase inhibitors. Regarding treatment, surgical resection is usually curative in localized BCs and adjuvant treatment is not routinely recommended. Multiple options for systemic therapy exist for patients with advanced BCs, although limited by a heterogeneity in the scientific evidence behind their use recommendation. These options include somatostatin analogues, everolimus, peptide receptor radionuclide therapy, chemotherapy, radiotherapy, antiangiogenic agents, and immunotherapy. In this article, we provide a comprehensive review about the molecular and genetic background of BCs, and about the treatment of local and metastatic disease, as well as the main paraneoplastic syndromes that have been associated with this tumor.

## 1. Introduction

Neuroendocrine tumors (NETs) of the lung comprise a heterogenous group of tumors, ranging from well-differentiated bronchial carcinoids (BCs) to highly malignant and poorly differentiated small cell lung cancer (SCLC) and large cell neuroendocrine carcinoma (LCNEC). On the one hand, neuroendocrine carcinomas (NECs), i.e., SCLC and LCNEC, are clinicopathological different entities than BCs, as they grow very rapidly and occur more frequently in patients with a history of smoking [1]. On the other hand, BCs are uncommon and usually slow growing neuroendocrine epithelial malignancies that occur frequently in never-smokers representing less than 2% of all lung cancers. However, there has been an increasing prevalence of BCs over the last 30 years, around 6% per year [2]. BCs can be subdivided in typical carcinoids (TCs) or atypical carcinoids (ACs), the latter ones being very rare (about 0.2%) [3]. TCs are slow growing tumors that rarely spread beyond the lungs, while ACs are more aggressive tumors and have a greater chance of metastasizing to other tissues [4].

Although most of the cases of BCs are sporadic, up to 5% of patients with multiple endocrine neoplasia type 1 (MEN1) harbor BCs, usually TCs [5]. Moreover, they may arise in the setting of a rare hereditary disease entity, the familial pulmonary carcinoid tumor [6]. It is known that NECs and BCs are different molecular entities, as comparative genomic hybridization studies and gene expression profiling data indicate separate clustering of BCs and NECs [3]. In this sense, chromosomal aberrations are more frequent in NECs than BCs, with the exception of the deletion of 11q, which is involved in the whole spectrum of lung NETs. Moreover, there are some molecular and genetic alterations in BCs that have been associated with survival outcomes and potentially with the response to therapy [7,8,9]. Overall, understanding the molecular background of BC is a key point in order to know the prognosis and to guide clinical decisions in metastatic settings.

Regarding treatment, most BCs can be cured by surgery [4]. The surgical approach is dependent on the size, location, and tissue type. The treatment of choice for centrally localized TCs is conservative resection (i.e., sleeve resection, segmentectomy or wedge resection), while for ACs, especially in a peripheral location, often anatomic resection (i.e., bi/lobectomy or pneumonectomy) is required [10]. In cases of centrally located intraluminal carcinoids, bronchoscopic excision may be also a therapeutic option [11]. Nevertheless, in advanced BCs, systemic therapies should be considered. Somatostatin analogues (SSAs) generally represent the first line treatment in advanced BCs and for symptomatic control of carcinoid syndrome [12,13,14]. Peptide receptor radionuclide therapy (PRRT) is a treatment option for selected patients with advanced and/or metastatic BCs that are somatostatin receptors (SSTR) positive at the imaging and show disease progression while receiving SSAs [15]. Standard chemotherapy regimens are usually of limited efficacy in BCs given their low proliferative capacity [16]. Targeted therapy including everolimus is the only approved drug in BC according to a phase III clinical trial in non-functioning extra-pancreatic NETs. [17]. The efficacy of other treatments such as antiangiogenic agents and immunotherapy, although promising, should be considered experimental [18,19,20,21].

In this article, we provide a comprehensive review on the molecular and genetic background of TCs and ACs, focusing on their differences with NEC, and potential prognostic and predictive biomarkers. Moreover, an overview on the treatment of local and metastatic disease, as well as the main paraneoplastic syndromes associated with BCs, including carcinoid syndrome, ectopic Cushing syndrome (ECS), and syndrome of inappropriate antidiuretic hormone (SIADH), are described.

## 2. Molecular and Genetic Background of Bronchial Carcinoids

Lung NETs encompass four histologic subtypes whose terminology and defining criteria have been endorsed over the last three classifications by the World Health Organization (WHO). This classification differentiates TCs, ACs, LCNEC, and SCLC [22,23]. The diagnosis of these four different types of lung NETs is mainly based on the mitotic count per 2 mm^2^, the presence of necrosis, and a wide constellation of cellular and architectural features fulfilling NET morphology [24]. Ki67 index is used to grade gastroenteropancreatic (GEP)-NETs, but it is not yet validated by the WHO for lung NETs. Molecular studies have supported the fact that NETs and NECs are different entities, but no causative relationship between NETs and NECs has been found and no mixed lesions between well and poorly differentiated subtypes have been described.

### 2.1. Differential Molecular Phenotype of Lung Neuroendocrine Carcinomas and Bronchial Carcinoids

Differences in the extent of molecular alterations between NECs and BCs may underline the differences in the aggressiveness [3]. Chromosomal alterations are much more frequent in NECs, except for the deletion of 11q, which is involved in the whole spectrum of neuroendocrine lung tumors. Smoking habits may explain part of these differences, as benzo[a]pyrene, a carcinogen of tobacco, can induce a DNA synthesis block leading to aberrant centrosome amplification, enhancing chromosomal instability [25]. Deletions of chromosome 3p are a hallmark of the high-grade pulmonary NEC. This deletion has been suggested to be one of the earliest events in NECs and is highly related to the smoking habit [26], whereas mutations of the *MEN1* gene (11q13) are typical from BC [27,28]. Deletion of 11q is the only chromosomal aberration for which the frequency of loss is comparable in carcinoids (28%) and NECs (32%) [3]. *TP53* and *retinoblastoma* (*RB*) mutations drive the pathogenesis of NECs but are considered a rare event in BCs [29]. G:C > A:T transitions, which are characteristic of smoking exposure, are less frequent in BCs as compared with NECs [30].

### 2.2. Molecular Alterations and Epigenetic Changes in Bronchial Carcinoids

Lung NETs encompass a heterogeneous spectrum of molecular alterations. Although ACs and TCs are clinically and histologically different entities, little is known about the factors underlying the prognosis for BCs and studies focused on genetic and molecular differences between both entities are scarce. Low mean somatic mutation burden (around 0.4 mutations/Mb) is a common feature of TCs. Although ACs have a similar set of mutations as TCs, they show a higher rate of mutations for each altered gene than TCs [31]. Here, we described the most common molecular alterations found in BCs, and the differences between TCs and ACs (Table 1 and Figure 1).

Covalent histone modifiers and subunits of the *SWI/SNF complex* are mutated in 40% and 22.2% of the cases of TCs and ACs, respectively, with *MEN1*, *PSIP1*, and *ARID1A* being recurrently affected [32]. Similar rates of mutations have been described in other studies [33]. Overall, *MEN1* is the most frequently somatically mutated gene in lung NETs (identified in 11–22% of BCs). In the group of lung NETs, *MEN1* alterations were almost exclusive to BCs [27,28]. The allelic loss of the *MEN1* locus is more common than *MEN1* mutations, and they are present in approximately 36% of sporadic BCs [34]. *MEN1* mutations derive in a loss of the protein function [7]. *MEN1* deletion is more frequent in ACs (70%) than in TCs (47%) [7,35]. Menin, the protein encoded by the *MEN1* gene, regulates gene transcription, cell proliferation, apoptosis, and genomic stability. Menin interacts with β-catenin, which acts as a transcription factor and whose dysregulation has been associated with the development and progression of many solid tumors, including several types of endocrine tumors [36]. In this sense, disarrayed E-cadherin or beta-catenin pattern was an independent predictor of lymph node metastases in patients with ACs [36], and *MEN1* gene mutations and deletions and decreased gene expression were associated with a poor prognosis in BCs, but were not related to sex, age, and smoking status [7]. Thus, *MEN1* mutation could represent a prognostic molecular biomarker in these neoplasms. Other genes belonging to the chromatin remodeling pathway include those of the *KMT2 (MLL)* family of covalent histone modifiers whose mutation rate is up to 14% in lung NETs [27].

**Table 1 cancers-14-00520-t001:** Molecular and genetic alterations in bronchial carcinoids.

Molecular/Genetic Alteration	Description of the Findings in Bronchial Carcinoids
Mutations in chromatin-remodeling genes	Covalent histone modifiers and subunits of the *SWI/SNF* complex are mutated in 40% and 22.2% of BC, respectively [32]. Allelic losses of the *MEN1* locus in 36% of sporadic BC
Somatic mutational burden	BC showed a lower mean number of mutations (TC, 0.7; AC, 1.8) than carcinomas (LCNEC, 4.6; SCLC, 5.8) [31]
Loss of heterozygosity (LOH)	LOH at 5q21 in 0% of TCs and 25% of ACs [37] LOH of 17p13 in 45% of ACs and 10% of TCs [38]. LOH of 3p in 40% of TCs, 73% of ACs [39]. LOH ATRX protein in 20% of BC [40], more common in AC than in TC.
Chromosomal instability (CIN)	CIN is increased in metastasized vs. non-metastasized carcinoids (gains 71% vs. 51%, losses 76% vs. 51%, cases with no chromosomal alterations 14% vs. 31%, respectively) [41]
Mutations in PIK3CA	Mutations in exon 9 and 20 in 13% of TCs and 39% of ACs [10]
Gene expression profiling	Upregulated genes: *RET, ASPM, BIRC5, BUB1, CEP55, FANCA* [42] Downregulated genes: *OTP, PCK1, ASB4, FOLRI1, CD44* [42,43].
Epigenetic changes	Promoter hypermethylation: *RASSF1A* [44] and *P15INK4b* [45] Histone modifications: Downregulation of *H4KM20* and *H4KA16* [46] miRNA upregulation: miR-129, -323-3p, -487b, -410, -369-3p, 376a [47] miRNA downregulation: miR-203, -224, -155, -302, -34b, -181b, -193a, -5p, -34b

AC: atypical carcinoid; BC: bronchial carcinoid; LCNEC: large cell neuroendocrine carcinoma; PIK3CA: phosphatidylinositol-4,5-bisphosphate 3-kinase catalytic subunit alpha; SCLC: small cell lung cancer; TC: typical carcinoid.

Loss of heterozygosity (LOH) at 5q21 is correlated with poor survival in BCs, showing a significant difference in LOH frequency between TCs (0%) and ACs (25%) [37]. Furthermore, a meta-analysis revealed that LOH at chromosome 5q in lung NECs was higher, at 75%, than in BCs in which the frequency was 1.4% [48]. Regarding the LOH and point mutations of the *TP53* locus on chromosome 17p13, they have been detected more frequently in ACs (45%) than in TCs (10%), and their prevalence increases with the severity of the tumor type [38]. LOH at chromosome 3p is the most frequent change in lung NETs and has been observed in 40% of TCs, 73% of ACs, 83% of LCNEC, and 85% of SCLC [39]. Loss of alpha thalassemia/mental retardation syndrome X-linked (ATRX) protein, that is part of the SWItch/Sucrose Non-Fermentable (SWI/SNF) family of chromatin remodeling proteins, is present in 20% of BCs. Mutations in *ATRX* result in the loss of the nuclear ATRX protein, which leads to the activation of the alternative lengthening of telomeres pathway, resulting in chromosomal instability (CIN), tumor heterogeneity and metastases [40]. In this sense, *ATRX* mutations were more frequently detected in ACs than TCs and it was associated with a shorter disease-specific survival (HR = 11, 95% CI, 1.8–68, *p* = 0.01), after adjusting for lympho-vascular invasion and presence of metastatic disease at time of diagnosis [8].

CIN, in terms of increased frequency and extent of chromosomal alterations, was found as a common event in atypical and metastatic BCs [41]. This fact is also supported by Warth et al. [41]. They reported that the frequency of CIN is significantly increased in metastasized versus non-metastasized BCs (gains 71% vs. 51%, losses 76% vs. 51%, cases with no chromosomal alterations 14% vs. 31%, respectively) [41].

Recurrent mutations in the kinase domain of *PIK3CA* (exon 9 and 20) have been described in 13% of TCs and 39% of ACs by Sanger sequencing [9]. In this study, the frequency of *PIK3CA* gene mutations increased with the biological aggressiveness of the lung NETs. Nevertheless, another more recently published study found lower prevalence rates of mutations in genes belonging to the PI3K/AKT/mTOR pathway, with a prevalence of 0% in TCs, 1% in ACs, and 3% in NEC [27]. Multiple drugs can be used to inhibit the PI3K/AKT/mTOR pathway, including everolimus, the only approved drug in advanced BCs based on the results of a phase III trial [17]. In addition, certain studies have found that *mTOR* expression and gene mutation can predict the efficacy of mTOR inhibitors [49,50].

Regarding gene expression profiling, in poor prognosis BCs, a set of genes was found to be upregulated, including the proto-oncogene *RET* and other genes involved in cell cycle control, such as *ASPM, BIRC5, BUB1, CEP55, FANCA*, and others, whereas *OTP, PCK1, ASB4, FOLRI1, CD44*, and others were downregulated [42]. The prognostic value of the *OTP (orthopedia homeobox)* gene and the stem cell marker CD44 (cell adhesion and migration) (neuroendocrine cell development) has been validated in larger series of BCs, confirming that their loss of expression is associated with poor prognosis [43]. *BIRC5, BUB1, CD44, IL20RA*, and *KLK12* were also independent predictors of patient outcome [42] (Table 1).

In lung NETs, only a limited number of studies analyzing global DNA methylation and chromatin remodeling patterns across different histological subtypes have been reported. One of the epigenetic molecular mechanisms described in BCs include promoter hypermethylation of *RAS-association domain family 1 (RASSF1)* gene [51]. A hypermutated profile was detected in two BCs carrying *DNA polymerase theta gene (POLQ)* mutations, which are known to make cells prone to DNA double-strand breaks and homologous recombination [27]. ACs have a higher frequency (71%) of *RASSF1A* methylation than TCs (45%). However, methylation of the *RASSF1A, p16, APC* and *CDH13* genes was significantly higher in NECs than in BCs [52,53]. The nuclear overexpression of protein arginine methyltransferase-5 (PRMT5) was negatively associated with tumor grade, thus reflecting potential differences in the epigenetic control of oncogenesis of low- and high-grade lung NETs [54]. Furthermore, methylation levels of the *MCAM* gene were demonstrated to be significantly higher in BCs than in other neuroendocrine neoplasms, supporting its role as a site-specific molecular biomarker [55]. These methylation pattern differences between lung TCs, ACs, SCLCs, and NSCLCs reinforce the fact that these tumors differ in tumorigenic process involving different pathways [38] (Table 1 and Figure 1).

The expression of microRNA (miRNAs) profiles has also been associated with survival in lung NETs. miRNAs are negative endogenous gene-expression regulators by binding complementary sequences in target mRNAs, resulting in their selective degradation or selective steric inhibition of translation [56]. A recent study found that miR375-3p levels were higher in low-grade lung NETs samples compared to non-neuroendocrine lung tumors. However, miR375-3p was not able to distinguish among different types of lung NETs [57]. Another study found 20 miRNAs that were differentially expressed in BCs compared with NECs; 14 of them were upregulated and 6 were downregulated in BC. However, they did not find any association between miRNAs profile and the risk of lymph node metastasis [58]. Nevertheless, other study analyzed 752 miRNAs in a discovery set of 12 BCs, and found that miR-409-3p, miR-409-5p and miR-431-5p were significantly downregulated in lymph node metastatic carcinoid tumors [56,57,58,59] (Table 1 and Figure 1).

## 3. Treatment of Local Disease

### 3.1. Surgery

Several aspects have to be considered related to the surgical treatment of BCs. Among them: (1) principles of surgical treatment in local disease, and extent of surgical resection, (2) main differences in management and results between TCs and ACs, (3) incidence and prognostic significance of carcinoid lymph node metastases, and (4) prognostic significance of pulmonary multifocal neuroendocrine proliferation, with TCs. Patients with TCs and ACs who are candidates for lung resection should undergo pulmonary function testing to help determine surgical risk [59]. Regarding biopsy in central BCs, in patients at high risk for bleeding, rigid bronchoscopy may be preferred than flexible bronchoscopy [4]. For peripheral BCs, biopsy specimen may be obtained by endoscopic transbronchial biopsy or transthoracic CT-guided biopsy [4]. Hemorrhage has been described in up to 27% of bronchoscopic biopsy in BCs, but severe bleeding is rarely encountered (2%) [60]. Nevertheless, other series reported an incidence of moderate to severe hemorrhage associated with bronchoscopic biopsy as high as 52%, but also in this series, only one patient required blood transfusion [61]. Surgical resection is the treatment of choice for localized BCs. Surgical resection of lymph node-negative BCs is associated with a survival advantage over nonoperative treatment. However, the disease specific survival at 5 years was still high without any treatment, suggesting that observation of asymptomatic peripheral TCs or endoscopic management of symptomatic central BCs may be considered in patients at high risk for surgical resection [62].

The goal of surgery is to conserve as much of the normal lung tissue as possible while performing the resection with a tumor-free resection margin (R0), which is associated with good prognosis. Patients with small peripheral BCs may be candidates for minimally invasive access including lung resection and lymph node dissection (usually a video-assisted lobectomy/segmentectomy). The surgical access for centrally located BCs or those with suspected/proven metastatic lymph nodes, is usually by thoracotomy approach. In a SEER database review, lobectomy (51.2%) was the most common surgical approach, while another major approach was sublobar resection with wedge resection or segmentectomy (24.1%) followed by ablation, pneumonectomy, bronchoplasty, or extended resection [63]. The best evidence available at this moment states that in the case of localized disease, the surgical techniques of choice are lobectomy or sleeve resection with systematic nodal dissection. Sublobar resection is an acceptable alternative if complete (R0) resection can be achieved in peripheral <2 cm TCs (T1aN0M0). For central TCs or ACs, where possible, bronchial sleeve resection (no lung tissue is removed) or a sleeve lobectomy should be carried out rather than pneumonectomy (with intraoperative frozen section of the resection margins) [4,59,64].

The average 5-year survival rate in the reviewed literature for TCs and ACs is 93% (range 88–97%) and 69% (range 40–86%), respectively [65]. Current standard for staging is the UICC/AJCC 8th edition for TNM staging system for lung cancer; however, has not yet been validated for BCs. Yoon et al. [66] showed a combined stage HR confidence interval and subcategories of the stage groups that demonstrated substantial overlap and did not show significant differences in disease specific survival. These findings may limit the usefulness of the TNM staging system, particularly in stages II and III. Many prognostic factors have been described [66,67]. ACs are associated with higher recurrence rates and anatomical lung resection should be preferred. Lymph node involvement and their prognostic impact is also well documented and increases in AC histology [68].

### 3.2. Endobronchial Resection

TCs and ACs without an extraluminal component can be treated at bronchoscopy resulting in an excellent long-term outcome with more tissue-sparing than immediate surgical resection. Cryotherapy and endoluminal and laser bronchoscopy may be a curative option and offers many advantages as it is rapid, immediately effective, and repeatable. Laser bronchoscopy may also be used in association with other therapies (i.e., SBRT) in cases of widespread intramural infiltration with an extraluminal component [69].

Current international guidelines of lung NETs propose that endobronchial TCs or ACs without an extraluminal component and with a small base attachment (<1.5 cm^2^) can be treated via endobronchial resection, resulting in an excellent long-term outcome and more tissue-sparing than surgical resection [59]. Bronchoscopic removal of endobronchial lesions may also reduce the risk of post-obstructive infections and improve pulmonary function, allowing the patient to undergo surgery in better clinical and respiratory state, so it can be used as a bridge to surgical resection [70]. This treatment strategy represents a minimally invasive and parenchyma sparing alternative, by identifying the accuracy of margins and extension of tumor. Such local therapies need to be considered in the context of patient medical status, type of BC, and robust imaging including functional imaging to exclude nodal spread [59]. Some authors suggest that initial bronchoscopic treatment strategy may be a justifiable alternative for surgical resection in 42% of the BC and needs to be implemented in the treatment of these patients [71].

### 3.3. Other Treatments in Local Disease

The experience with stereotatic beam radiation therapy (SBRT) for BCs is much more limited, being reported in only a few small series and no solid evidence support their use [72]. It seems reasonable to expect that tumor control rates with cryotherapy and radiofrequency ablation would be similar to other lung lesions. Ablative therapies may be particularly suitable for localized TC or AC lesions in patients unfit for or declining surgery [59].

## 4. Treatment of Advanced Disease

Due to the low incidence of these tumors, the management of the advanced disease must incorporate a multidisciplinary approach in reference centers to improve the therapeutic options and optimize the research in BCs. In addition, related to the above, few data are currently available regarding treatment for relapsed or advanced disease and the optimal sequencing algorithm is not yet stablished. Therefore, an individualized treatment approach should be carried out [4,16,73,74] (Figure 2 and Table 2). Regarding high-grade pulmonary NECs, in patients who have limited stage, without hilar or mediastinal nodal involvement (TNM stages I-IIA), may be considered for resection of the primary tumor with mediastinal lymph node sampling or dissection [75]. In patients with metastatic high-grade pulmonary NECs, first-line treatment generally consists of platinum-based chemotherapy with response rates that range from 42% to 67% and a median survival of 15 to 19 months [76]. Temozolomide with or without capecitabine and bevacizumab is generally given as second line therapy [77]. Topotecan has also been recommended in second line based on extrapolation from treatment of SCLC. Moreover, clinical trials with sunitinib and everolimus are reportedly under way [76].

### 4.1. Somatostatin Analogues

For patients who have slowly progressive and somatostatin-receptor-positive BCs as determined by somatostatin-receptor-based diagnostic imaging (based either on Gallium-68 PET or DOTATATE PET imaging), therapy with somatostatin analogues (SSAs) can be considered [78,79]. Moreover, SSAs constitute the gold standard for symptomatic relief.

SSAs can induce stabilization in 30–70% of patients with well-differentiated NETs as demonstrated, mainly, in retrospective series and phase II single arm studies [80,81,82]. SSAs have demonstrated to inhibit tumor growth in NETs [12,83], but their use as monotherapy in BCs that express SSTR has been based on small studies and data coming from GEP-NETs, such as the PROMID and CLARINET trials [12,84,85]. Two main retrospective series including patients treated with first line SSAs, showed a disease control rate (DCR) of approximately 80% with a median progression free survival (PFS) between 17.4 and 28.6 months [85,86]. Greater benefit was identified in patients with Ki67 ≤ 10% and TC histology with controversial results according to functioning status. The largest prospective study, although closed early due to slow accrual, is the phase III SPINET trial (NCT02683941) [87] (Table 2). The 51 patients randomized to lanreotide depot/autogel achieved a median PFS of 16.6 months, during double-blind and open label phases. According to the histologic subtype, lanreotide achieved a greater benefit in patients with TCs (*n* = 29) with a median PFS of 21.9 months compared with those with ACs (*n* = 22) that reached a median PFS of 14.1 months. The use of SSAs in combination with other targeted agents in BCs has been prospectively analyzed in two main clinical trials (Table 2). The phase II LUNA trial with 3-arms of treatment included patients with lung and thymus NETs [88]. The results showed promising data on the activity of SSAs in combination with everolimus in this group of patients, but more powered trials are needed to support this combination strategy. The ATLANT study [88] is a prospective single-arm study of lanreotide autogel and temozolomide in 40 patients with lung and thymic NETs. The promising results obtained in activity and safety require a control arm and greater sample size in further trials.

### 4.2. Treatment of Liver-Dominant Metastatic Disease

For patients who have limited, potentially resectable liver metastasis, surgical resection is recommended [93,94]. Although most cases of advanced disease will not be cured by surgery, symptoms of hormone hypersecretion are effectively palliated, and prolonged survival is often possible. Other liver-directed therapies for hepatic-predominant disease include hepatic artery embolization (bland particle embolization, chemoembolization, radioembolization) and radiofrequency ablation, although few studies are available on the efficacy of these modalities in patients with BCs [95,96]. The added value of combination of locoregional therapies as a complement to surgery or systemic therapy in case of progressive disease should be considered [93,94].

### 4.3. Targeted Therapy

Mammalian target of Rapamycin (mTOR) has been identified as a relevant kinase activated in the PI3K signaling pathway of BCs [97], and mutations in this pathway contributes to tumorigenesis. Everolimus is the first targeted agent to show robust anti-tumor activity with acceptable tolerability across a broad range of NETs, including those arising from the lung. The RADIANT-4 trial, a phase III study included 302 patients with advanced, progressive, non-functional, lung (30%) or gastrointestinal NETs. Patients were randomized to receive everolimus 10 mg/day or placebo [98]. Everolimus was associated with a significant improvement in PFS in the intention to treat population, the primary endpoint of the study [median PFS = 11.0 months for everolimus versus 3.9 months for placebo (0.48 [95% CI 0.35–0.67], *p* < 0.00001)]. Based on these results, everolimus was approved in February 2016 by the US Food and Drug Administration (FDA) for the treatment of adults with progressive, well-differentiated, nonfunctional lung NETs with unresectable locally advanced or metastatic disease. According to the ESMO guidelines, everolimus is recommended in patients with metastatic ACs or in TCs with significantly progressive disease or as a subsequent treatment line after SSA therapy failure [99]. The RADIANT-2 trial [100] also included patients with lung NETs, but no stratification according to the primary site origin was performed. In a similar manner as occurred in the overall population of this study, the subgroup analysis in lung NETs showed a clinically meaningful benefit in median PFS with an increase of 8 months for the addition of everolimus, but without statistical significance (*p* = 0.228).

Treatment with antiangiogenic agents in BCs has also been analyzed based on their highly vascular biology. Sunitinib, an orally administered kinase inhibitor, has activity against different tyrosine kinase including VEGFR-1, -2, -3, platelet-derived growth factor receptor (PDGFR)-a, and -b [101]. A phase II study was conducted in 109 patients with NETs of whom 14 patients had foregut origin, including BCs. The overall response rate (ORR) in the group of patients with extra-pancreatic NETs was 2.4%, with a stable disease rate of 83% and a median time to tumor progression of 10.2 months and one-year survival rate of 83.4% [19]. The activity of pazopanib after failure to previous systemic treatments in progressive metastatic NETs showed some clinical benefit, including patients with BCs, in the PAZONET study. For the overall population (*n* = 44), the median PFS was 9.5 months, but for patients with lung or thymus NETs (*n* = 8), the median PFS was 3.4 month (significantly lower in comparison to the group of patients with pancreatic origin that achieved a median PFS of 12.8 months [18]. Bevacizumab is an anti-vascular endothelial growth factor (VEGF) monoclonal antibody whose role in NETs treatment has also been studied. Very promising results were identified in a phase II study that randomized patients to bevacizumab plus octreotide or pegylated interferon alpha-2b plus octreotide with an ORR of 18% (*n* = 4) and stable disease of 77% (*n* = 17) for the bevacizumab arm [20]. Those results led the phase III trial (SWOG S0518) that included 427 patients but did not find a significant difference between both arms in terms of median PFS (16.6 months for the bevacizumab arm and 15.4 months for the interferon arm; HR 0.93; 95% CI, 0.73 to 1.18; *p* = 0.55) [102]. No subgroup analysis was included according to the primary tumor origin. The combination of sorafenib plus bevacizumab was analyzed in a phase II study of 44 NETs, including 19 patients with foregut NETs, showing a clinical benefit (ORR of 9.4%), but unfavorable safety results compared with both drugs given as monotherapy [103]. Overall, many small trials have tried to assess the role of antiangiogenic agents in lung NETs, but larger phase III trials lack showing the benefit of this therapeutic strategy. In this sense, in the last two years, two other antiangiogenic drugs, axitinib or surufatinib have been assessed in phase III trials including this group of patients. Firstly, the SANET-ep study [91], only conducted in China, analyzed the efficacy and safety of surufatinib in patients with extra-pancreatic NETs, including 23 patients with BCs (11.6%). The study was terminated early according to the results achieved in the interim analysis; the investigator-assessed median PFS was 9.2 months (95% CI 7.4–11.1) in the surufatinib group versus 3.8 months (3.7–5.7) in the placebo group (HR 0.33; 95% CI 0.22–0.50; *p* < 0.0001). The efficacy of surufatinib in the US population was assessed in pancreatic NETs and in the European population is currently under research in an open-label phase 2 study (NCT04579679) with different cohorts of patients (Cohort A: NETs of lung origin; Cohort B: NETs of small bowel origin; Cohort C: NETs of non-small bowel, non-pancreas, and non-lung origin; Cohort D: NETs of any origin, DDI substudy) [104] (Table 3). The role of axitinib has been assessed in the AXINET phase II/III study [92] conducted in patients with advanced G1-G2 non-pancreatic NETs, including 71 patients with lung NETs. Patients were randomized (1:1) to receive octreotide LAR with axitinib or placebo. In this study, axitinib in combination with octreotide LAR showed a clinically meaningful greater ORR than in the placebo group (17.5% vs. 3.8%, *p* = 0.0004), with a tolerable safety profile. Unfortunately, the primary endpoint of this trial, PFS per investigator assessment, did not reach statistical significance (median PFS 17.2 month in axitinib arm vs. 12.3 months in place group, HR 0.816, *p* = 0.169). An update was presented at the ESMO Congress 2021 showing the results of the secondary endpoint added when the trial changed to a phase III multicenter international study. This endpoint was PFS by central blinded assessment and showed a significant difference between the group of axitinib compared with placebo (median PFS was 16.6 months vs. 9.9 months; HR0.71, *p* = 0.017) [105].

Finally, other pathways inhibited by targeted agents are currently under research including epidermal growth factor receptor (EGFR) inhibitors, fibroblast growth factor (FGFR), or MET receptor (Table 3).

### 4.4. Peptide Receptor Radionuclide Therapy (PRRT)

PRRT have also been assessed in patients with BCs. Using the radiolabeled SSA Lutetium, 177Lu-DOTATATE is another appropriate treatment option and the PRRT recommended for patients with advanced, progressive, and SSTR-positive tumors according to clinical guidelines [106]. The NETTER-1 trial showed the benefits of 177Lu-DOTATATE in patients with advanced midgut NETs, demonstrating a markedly longer PFS and a significantly higher response rate than high-dose long-acting octreotide [107]. Although the experience of 177Lu-DOTATATE is more limited in BCs, high rates of antitumor activity has been reported in several studies [108,109,110,111,112,113]. Ianniello et al. [113] showed in one of the largest series including 34 patients with BCs treated with 177Lu-DOTATATE, a DCR of 80% (6% complete response, 27% partial response and 47% stable disease) and a median PFS of 20 months. In addition, Brabander et al. [114] demonstrated the efficacy of this PRRT in a large cohort of Dutch patients, of whom 23 had a BCs. The ORR was 30%, and an additional 30% of patients had stable disease. At a median follow-up of 78 months, the median PFS was 20 months, and the median OS was 52 months. Long-term toxicity included acute leukemia in 4 patients (0.7%) and myelodysplastic syndrome in 9 patients (1.5%). There was no therapy-related long-term renal or hepatic failure. The most recent retrospective report only focused on the activity of PRRT in lung NETs from two ENETS Centers of Excellence, included 48 patients (*n* = 43 ACs and *n* = 5 TCs) [115]. Results were also promising with an ORR according to RECIST criteria reached 20% and stable disease was identified in 44% of patients. The median PFS and overall survival (OS) were 23 months and 59 months, respectively. 177Lu-DOTATATE was approved by the US FDA and EMA in January 2018 for the treatment of somatostatin receptor-positive GEP-NETs in adults. Although the approval did not cover BCs, off-label use can be considered in selected patients with somatostatin receptor-expressing tumors and failure to previous effective therapies. However, the optimal place in the therapeutic algorithm of PRRTs is still under debate, as greater prospective trials are needed.

Other radiolabeled SSA evaluated in BCs is 90Y-DOTATOC, another well-tolerated treatment for NETs with a remarkable ORR, prolonged survival time, and symptomatic response [112,116]. A large study of 1109 patients with metastatic NETs included 84 BCs, of which 28.6% showed a morphological response as estimated by RECIST criteria and 38.1% showed a clinical response with a mean survival of 40 months. Grade 3–4 transient hematologic toxicities were reported in 12.8% of patients and grade 4–5 permanent renal toxicity in 9.2% [112].

### 4.5. Chemotherapy

There are multiple cytotoxic drugs that have been analyzed for advanced lung NETs as monotherapy or in combination. Results should be interpreted with caution due to the retrospective design, small number of patients included and mixed population of primary tumors, treatment lines and grades, as well as the lack of central pathology review. In general, response rates to carboplatin/cisplatin-based therapies are low in well differentiated lung NETs [117]. However, better results have been obtained with other chemotherapy regimens, such as temozolomide/streptozotocin (STZ) or oxaliplatin based chemotherapy. Thus, temozolomide monotherapy or in combination has been evaluated in retrospective and single arm phase II studies achieving a DCR of 52% and median PFS between 5.3 and 7.3 months [118]. Based on the results in pancreatic NETs, a recent retrospective study showed very interesting results from the combination of temozolomide and capecitabine in terms of ORR 30% and DCR of 85% administered in 20 patients with lung NETs (*n* = 14 TCs, *n* = 5 ATs, *n* = 1 Large-cell NECs) with a good safety profile [119]. Data with STZ come from a randomized trial analyzing 5-FU–STZ versus 5-FU–doxorubicin in symptomatic carcinoids including BCs, 5-FU–STZ combination improved survival compared with the doxorubicin-based regimen, suggesting that doxorubicin does not confer any benefit in these patients [120]. In another analysis of 5-FU, STZ and cisplatin in 79 patients with progressive NETs, of which 8 were lung NETs, the response rate for non-pancreatic primary sites was 25%. Among BCs, 2 patients showed partial response, and 4 patients stable disease [121]. Another trial with the combination of capecitabine with STZ found no added value by adding cisplatin [122]. Oxaliplatin based therapies have been analyzed in a series of 45 patients with TCs or ACs reaching an ORR of 20% and DCR of 84% with a median PFS of 15 months [123]. With a mixed population, other reports have shown an ORR of 12% (*n* = 5/42)–13% (*n* = 2/15) and DCR of 78% (*n* = 33/42)–80% (*n* = 12/15) [124,125].

Therefore, chemotherapy in patients with well differentiated lung NETs is recommended in refractory patients to previous effective treatment lines and in highly progressive tumors. Temozolomide based regimens are the recommended ones when the decision of chemotherapy administration is done. In particular cases of aggressive ACs, platinum-based chemotherapy can be considered. Nevertheless, the level of evidence supporting the use of these drugs in this setting is low, and further clinical trials with those and new strategies are needed.

### 4.6. Immunotherapy

There is little knowledge about the efficacy of immune checkpoint inhibitors (ICI) in patients with advanced BC. Anti-programmed cell death-1 (*anti-PD-1*)/antiprogrammed cell death ligand-1 (*anti-PD-L1*) ICI are well incorporated into treatment algorithms for advanced lung cancer based on the results of numerous randomized clinical trials demonstrating an OS benefit [99]. However, data regarding clinical activity of immunotherapy in advanced BCs is limited based on phase II trials with different primary tumors origin [126]. Two main trials evaluating the role of PD-1 inhibitors monotherapy, spartalizumab and pembrolizumab, in 30 and 16 patients with lung NETs, showed an ORR of 16% and 0% and a median PFS of 7% and 3.7%, respectively. Looking for an increase in tumor activity by the synergy of PD-1 and CTLA-4 inhibition, the DART trial evaluated the clinical activity of ipilimumab plus nivolumab in nonpancreatic NETs demonstrating an ORR of 44% in patients with high grade NECs [127]. No improvement in ORR was achieved in well differentiated NETs. The DUNE study [128], a multi-cohort phase II study of durvalumab plus tremelimumab for the treatment of patients with advanced and previously treated NETs of GEP or lung origin, found that the primary endpoint of clinical benefit rate at 9 months by RECIST v1.1 criteria for typical and atypical BCs (*n* = 27) was 7.4% (not achieved). Overall, more studies are needed to know the real efficacy of immunotherapy in the different NETs, with the adequate endpoints for ICI evaluation and the potential optimal position of this therapeutic strategy as monotherapy or in combination in the therapeutic algorithm of BCs.

### 4.7. Epigenetics

The role of epigenetic changes in BC tumor progression has been identified in different studies as explained above. In fact, as chromatin remodelers have a key role in gene expression or DNA replication, the development of drugs targeting these pathways is of great interest in current research of advanced solid and hematologic tumors. Particularly in NETs and of lung origin, a phase I trial (NCT02875223) has recently been published evaluating the role of CC-90011, a reversible inhibitor of LSD1 (Lysine-specific demethylase 1), that can act, through histone H3, as a transcriptional corepressor or coactivator through demethylation of lysine 4 or 9, respectively [129]. Considering the dose escalation and expansion phases, 34 patients with NETs and NECs were included and a prolonged stable disease was observed mainly in patients with well differentiated primary lung tumor origin (SD > 6 months in 8 patients). Due to these findings, the expansion cohort, including patients with BCs, is currently recruiting patients and opening a new therapeutic strategy in these NETs. Other histone deacetylase (HDAC) inhibitors, such as belinostat, are under research, but in high grade lung NEC.

## 5. Treatment of Paraneoplastic Syndromes

### 5.1. Carcinoid Syndrome

Carcinoid syndrome is the most frequent functioning syndrome in BCs. The clinical picture of carcinoid syndrome is induced by the release of tumor-derived bioactive peptides, predominantly serotonin, into the systemic circulation. The excessive amounts of circulating serotonin produce an increase of bowel movements, facial flushing, bronchoconstriction, and may result in fibrotic complications of mesentery and heart structures [130]. It is estimated that 20–40% of patients with carcinoid syndrome had right-sided valvular fibrosis leading to valve dysfunction and right heart failure [131]. Urinary 5-Hydroxyindole Acetic Acid (5-HIAA) is the most frequently used biochemical marker for the diagnosis and follow-up of carcinoid syndrome. The diagnostic accuracy of urinary 5-HIAA it is reported in approximately in 70% of sensitivity, and 90% of specificity [130]. For the diagnosis of carcinoid heart disease, cardiac imaging has a principal role for assessment and monitoring. The first-line imaging modality in carcinoid heart disease evaluation is two-dimensional transthoracic echocardiography (2D TTE), which offers both morphological and functional assessment [132].

SSAs constitute the gold standard for symptomatic control, improving the frequency of flushing symptoms and diarrhea. Both commercially available somatostatin analogs that predominantly acting on SSTR2, octreotide long-acting release (LAR) and lanreotide Autogel^®^ offer similar efficacy in terms of carcinoid syndrome symptom response [133]. The doses used in carcinoid syndrome are widely variable across the different studies, but a recent meta-analysis [133] found that decreasing the injection interval to 21 days or switching to the alternative SSA leads to a significantly reduction of symptoms. In patients with refractory symptoms, short-acting octreotide may also be needed as add-on rescue medication along with the long-acting formulations [133]. A study looking at the management of carcinoid syndrome in 7 patients with AC showed effective control of symptoms with short-acting octreotide [134]. In addition, the presence of carcinoid heart disease should be carefully monitored. Moreover, prophylaxis against carcinoid crisis should be carried out using a dose of SSA that achieved an adequate symptoms control before surgical or locoregional interventions [73].

In SSA-refractory patients or in cases with persistent symptoms of carcinoid syndrome despite treatment with an SSA, the serotonin synthesis inhibitor telotristat ethyl is an interesting therapeutic option based on the safety and efficacy results in the TELECAST phase 3 trial [135] and compromises the preferred second-line option. On the other hand, a recent study analyzed the effect of 177Lu-DOTATATE in patients with NETs and carcinoid syndrome treated for the purpose of symptom reduction. This study demonstrated that PRRT with 177Lu-DOTATATE effectively reduced diarrhea and flushing and can be considered for symptomatic treatment of carcinoid syndrome insufficiently controlled with SSA [136] (Figure 3).

### 5.2. Ectopic Cushing Syndrome

BCs have the potential to secrete various peptides or hormones that can lead to paraneoplastic syndromes, such as Ectopic Cushing syndrome (ECS). Symptoms of Cushing’s syndrome, including hypokalemia, proximal muscle weakness, peripheral edema, metabolic alkalosis and/or hyperglycemia, are present in 1–2% of patients with lung NET and the presence of this syndrome has a poor prognosis [137,138]. The ideal treatment of ECS is complete excision of the ACTH-secreting tumor that can be performed rapidly or after preoperative preparation using cortisol-lowering drugs. Cushing’s syndrome can be controlled with commonly used treatments, such as ketoconazole, metyrapone, etomidate, or mifepristone. Ketoconazole, administered at a daily dose of 600–800 mg, is the most widely used and effective treatment. Metyrapone is another therapeutic option in this setting, with a starting dose of 1 g/day and a maximum dose of 4.5 g/day. For ECS, SSAs can be of value [73]. In the absence of hormonal control, other treatments include locoregional therapies such as liver palliative surgery, transarterial chemoembolization (TACE) and radiofrequency ablation (RF), combination of SSA with interferon, and PRRT in selected patients [73,139]. Bilateral adrenalectomy should be mainly considered in patients with ECS and very severe Cushing’s syndrome, when steroidogenesis inhibitors are unavailable, ineffective, or poorly tolerated. Bilateral adrenalectomy is highly effective for hypercortisolism control, with an immediate effect [140] (Figure 3).

### 5.3. SIADH

Patients with BCs can develop the paraneoplastic syndrome of inappropriate antidiuretic hormone secretion (SIADH), due to the capable of producing peptide hormones, among which the antidiuretic hormone (ADH, arginine vasopressin) is one of the most frequent [141]. Paraneoplastic SIADH may result from ectopic ADH production or from other tumor-related mechanisms leading to increased pituitary ADH secretion, and is characterized by hyponatremia, serum hypoosmolality, urine hyperosmolality, and an inappropriately elevated ADH level. Successful treatment of the underlying tumor, including restricted fluid intake, will usually result in prompt disappearance of the paraneoplastic SIADH (Figure 3).

## 6. Conclusions

A better understanding of the genetic and molecular background of BCs would allow a better estimation of the risk of disease progression and the personalization of treatment in cases of advanced disease. In BCs, *MEN1* gene mutations and deletions and decreased gene expression have been associated with a poor prognosis. *ATRX* mutation has been also linked to a shorter disease-specific survival; the PI3K/AKT/mTOR pathway represents a targetable mutation with kinase inhibitors. Surgical resection is usually curative in localized BC. For advanced BCs, there are different options for systemic therapy, including SSAs, PRRT, chemotherapy, radiotherapy, everolimus, antiangiogenic agents, and immunotherapy. SSAs usually represents the first-line therapy in slowly progressive positive SSTR lung NETs, as they can induce stabilization in 30–70% of the cases. Everolimus and PRRT are appropriate options for patients with progressive, disseminated disease, including SSTR tumors for PRRT therapy or after SSAs. Systemic therapy with cytotoxic chemotherapy should be considered in highly progressive disease and/or refractory to other effective therapies, such as SSAs, everolimus, or PRRT. The treatment sequence are not clearly stablished by lacking of prospective clinical trials to compare different approaches. In this setting is a key stone: the patient selection in order to provide an individualized management to patients.

New treatment strategies are under research to improve the therapeutic options in BCs as well as to optimize the scientific evidence on the real benefit of the current available drugs to determine the optimal sequence in each patient. Some of these trials, with immunotherapy and drugs targeting chromatin remodelers, among others, are currently ongoing to solve these questions, as well as other unmet clinical needs, as the benefit of combinations rather than monotherapy sequencing, treatment duration, or the identification of predictive biomarkers. Therefore, the progressive improvement in the biology and molecular knowledge of BCs will help to offer our patients a comprehensive approach to treatment.

## Figures and Tables

**Figure 1 cancers-14-00520-f001:**
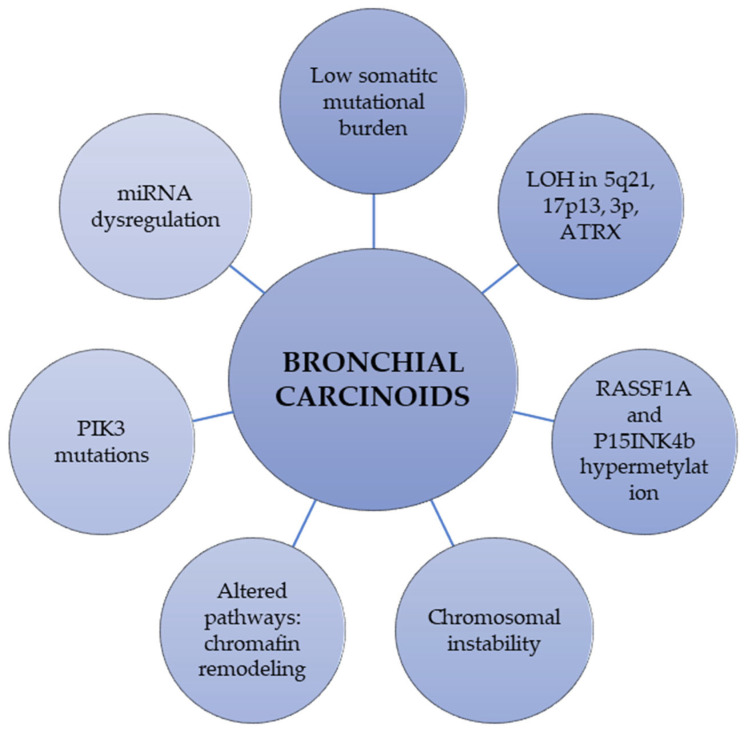
Overview of the most relevant molecular and genetic characteristics of bronchial carcinoids. ATRX: alpha thalassemia/mental retardation syndrome X-linked; LOH: loss of heterozygosity; RASSF1A: Ras association domain-containing protein 1; PIK3: phosphatidylinositol cinasa-3.

**Figure 2 cancers-14-00520-f002:**
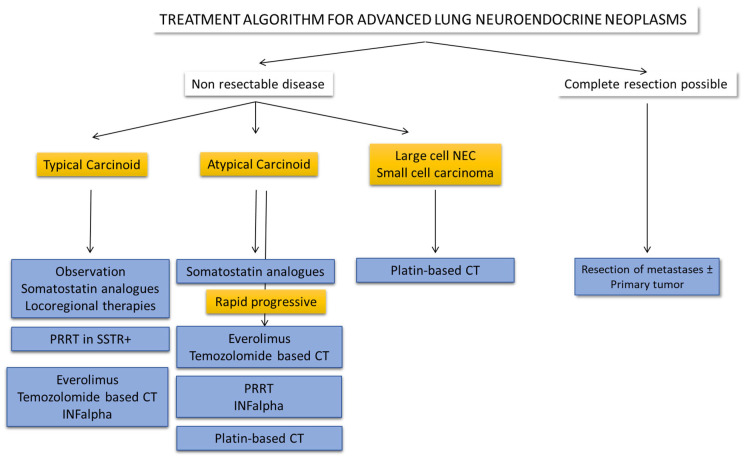
Treatment algorithm for advance lung neuroendocrine neoplasm. There is an unmet need for the classification of patients with a proliferative well differentiated NET with more than 10 mitosis/HPF.

**Figure 3 cancers-14-00520-f003:**
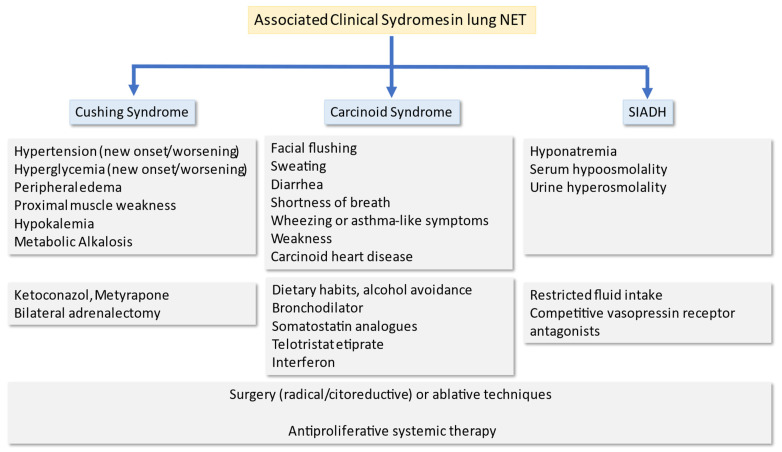
Associated clinical syndromes in bronchial carcinoids: clinical manifestation and treatment options.

**Table 2 cancers-14-00520-t002:** Results of the main prospective clinical trials in advanced bronchial carcinoids.

Trial	R	Previous Treatment	Number of Patients	Treatment	Comparator	Main Results
SPINET Phase III/ NCT02683941 [89]	R	First line	77 (early closed accrual)	Lanreotide autogel 120 mg/28 d	Placebo	mPFS = 16.6 (95% CI 12.8, 21.9) ORR = 14.0% vs. 0%
LUNA Phase II [14]	R	First line	124	Pasireotide 60 mg/28 d	Everolimus 10 mg/d Pasireotide 60 mg/28 d+ Everolimus 10 mg/d	PFS rate at 9 months = 39% (95% CI 24·2–55·5) vs. 33.3% (95%CI 19.6–49.5) vs. 58.5% (95% CI 42.1–73.7)
ATLANT Trial Phase II [88]	R	≤3 prior lines	40	Lanreotide autogel 120 mg/28 d and temozolomide 250 mg/m^2^ every 5 of 28 days	No control arm	DCR at 9 m = 35% (95% CI 20.63; 51.68)
Phase II	NR	SSA (84%), CT (38%), PRRT (26%)	34	177LuDOTATATE	-	DCR = 62% ORR = 15% mPFS = 18.5 m
RADIANT 4 Phase III/NCT01524783 [17]	R	SSA (53%); CT (26%); RT/PRRT (22%)	90 * (30%)	Everolimus 10 mg/day	Placebo	mPFS = 9.2 vs. 3.6 m (HR = 0.50)
RADIANT-2 Phase III/ NCT00412061 [90]	R	SSA (67%); CT (39%); IT (12%); TT (15%)	44 * (10%)	Everolimus 10 mg/day + Octreotide LAR	Placebo + Octreotide LAR	mPFS = 13.6 vs. 5.6 m (HR = 0.72)
EP SANET/ Phase III/ [91]	R	SSA (34%); CT (40%); Everolimus (8%)	23 * (11.6%)	Surufatinib 300 mg/day	Placebo	PFS * = 7.6 vs. 3.7 m (HR = 0.33)
AXINET/ Phase III/NCT [92]	R	SSA (48%); CT (14%); PRRT (4%); Everolimus (13%)	71 (28%)	Axitinib 5 mg/12 h + Octreotide LAR 30 mg/28 d	Placebo + Octreotide LAR 30 mg/28 d	mPFS = 14.36 vs. 11.72 (HR = 0.84)

*: number of patients with lung NET included in the trial; R: randomized; NR: not randomized; SSA: somatostatin analogues; CT: chemotherapy; Immunotherapy: IT; Targeted therapy: TT; PRRT: peptide receptor radionuclide therapy; RT: radiotherapy; DCR: disease control rate; ORR: overall response rate; mPFS: median progression free survival.

**Table 3 cancers-14-00520-t003:** Ongoing clinical trials in lung NET.

Study	Treatment	Comparator	*n*	Primary ENDPOINT	Secondary Endpoints	Reccruitment	NCT Identifier
Phase I/II	PEN-221	-	89	MTD	CBR, ORR	2016–2020	NCT02936323
NCI-2020-12905 Phase II	177LuDOTATATE	Everolimus	108	PFS	OS, ORR, safety	2021–2024	NCT04665739
PUTNET Phase II	177LuDOTATOC	-	50	ORR	PFS	2021–2022	NCT04276597
CABINET Phase III	Cabozantinib	Placebo	395	PFS	OS, ORR, safety	2018–2025	NCT03375320
CABOTEM Phase II	Cabozantinib + Temozolomide	-	35	ORR	PFS, CBR, OS	2021–2023	NCT04893785
2020-012-00EU1 Phase II	Surufatinib	-	76	DCR	PFS, DoR, Safety	2021–2022	NCT04579679
Phase II HMH008	Nivolumab + 177LuDOTATATE	-	30	ORR	ORR, PFS, Safety	2020–2020	NCT04525638
Phase II	Nivolumab + Temozolomide	-	55	ORR	Safety, PFS, OS	2018–2021	NCT03728361

MTD: Maximum Tolerated Drug; PFS: Progression Free Survival; CBR: Clinical Benefit Rate; DCR: disease control rate; OS: overall survival; ORR: overall response rate; NCT: number clinical trial.

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
