# Peer review of "Bronchial Carcinoids: From Molecular Background to Treatment Approach"

_cancers, 2022, doi:10.3390/cancers14030520_

Round 1

Reviewer 1 Report

My comments attached as word document and also pasted below:

Review for authors:

Brief summary: This manuscript is focused on bronchial carcinoids (BC), a type of lung neuroendocrine cancer. The review discusses these tumors from the molecular to the treatment level, with an emphasis on differences between BC and lung neuroendocrine carcinomas (NEC). Authors begin by discussing genetic mutations in both BC and NEC, such as MEN1/TP53/RB, and the chromosomal 11q deletion. Authors continue by describing molecular and genetic differences between the TC and AC subtypes of BC. The descriptions are comprehensive and include enough detail to allow readers to understand the importance of each molecular/genetic alteration, a strength of this manuscript. Authors do a good job referencing key studies and clinical trials in the field of lung NETs over the past few decades. Overall, I concluded authors support the concept that the various subtypes of lung NETs are molecularly distinct and progress uniquely, highlighting the importance of therapies directed towards specific tumor subtypes. This review is comprehensive and relevant to the field of lung NETs and provides an up-to-date discussion of molecular alterations and therapies.

General concept comments:

The manuscript has a nice progression from describing molecular level alterations to patient-level treatment options. The conclusion was shorter than I expected. Throughout the manuscript and within the conclusion, authors frequently emphasize the importance of finding new treatment options for patients with lung NETs. For example, the conclusion states: “New treatment strategies are under research…” It would be great if authors could expand on this statement in the conclusions section. For example, mentioning some of these new treatment strategies (with proper citations) would be helpful for readers to learn more about specific current research going on for lung NET therapies.

My main concern for the overall manuscript is flow of language and grammar, which I am confident authors can fix. There are multiple misspellings and the improper/inconsistent tense usage throughout the manuscript and within the figures (ex. “inestability” in Fig 1).

Specific comments: 

Line 100: To improve the relevance of the Ki67 sentence, I would suggest clarifying the importance of Ki67 index in the broader field of NETs (it is used to grade GEP-NETs, for example) before mentioning that is not currently recognized/validated by the WHO for diagnosing lung NETs.

Lines 18, 125,: Manuscript would flow smoother with less transitional wording. For example, on line 125 the phrase “However, AC, although has a similar set…” would read more clearly by removing either “however” or “although”.

Line 166: Please clarify: “….was found as a common event in atypical and BC [43]” . Atypical is a subtype of BC, therefore the “and” is confusing.

Lines 173-174: Please clarify which type of lung NET harbors lower prevalence rates of mutations in genes belonging to the PI3K/AKT/mTOR pathway.

Section 3.2. Endobronchial Resection: Ensure statements are properly cited. For example, the second paragraph in this section begins by saying “Recent studies”, but the only citation in this paragraph references the Commonwealth Neuroendocrine Tumour Research/NANETs 2020 guidelines update.

Line 323: Please clarify if indeed talking about advanced disease, the question mark is confusing: “(of advanced disease?)”

Sections 4.6 and 4.7: Authors begin using the NEN acronym, which was never previously defined. Please define NEN as neuroendocrine neoplasms.

Figure 2: The green box labeled “symptom control” is confusing to me. SSAs are used for symptom control…but the placement of this green box makes it seem like everything below it is used for symptom control, which is misleading in this figure, especially in the context of TCs.

Minor detail, but I suggest being consistent with your use of either “analog” or “analogue”.

Author Response

Thank you very much for your sound and constructive comments and for giving us the opportunity to review and improve our manuscript.

In the attached file you can find a detailed point by point answer to your comments and concerns.

Reviewer 2 Report

I think that there are some issues that must be discussed in a revision.

(1) What does consolidate mean?  lines 26 and 87.  Please consider revision

(2) The authors list the possible surgical approaches like a menu.  The reader does not know which is indicated.  For BC, what surgical procedure is recommended?  Please note that these tumors are centrally located

(3)  In diagnosis on BC, is a biopsy necessary?  This was not discussed but these tumors are very vascular and a biopsy has been associated with hemorrhage.  This needs to be mentioned to the reader.  Some simply recommend surgical resection and not biopsy.

(4)  What is the role of bronchoscopic excision and does a lymph node dissection need to be done?

(5)  The authors need to amplify the discussion of treatment for high-grade pulmonary NEC.  This is inadequately covered.

(6)  The initial work-up requires somatostatin-receptor-based diagnostic imaging (line 296).  Does this mean octreoscan or DOTATOC scan?   I should mean the latter because it is more sensitive than the former.  You should provide some recent references to support your recommendation.

(7) How do you diagnose Carcinoid syndrome?  Echo cardiogram to R/O right sided heart disease is recommended for all patients at diagnosis and every 5-10 years afterward.  The discussion of carcinoid crisis should be emphasized because this is a life-threatening complication and must be avoided.  What SSA should be used dose and how is it administered?  When in surgery?

(8) In ectopic Cushing syndrome, the authors fail to mention the role of bilateral adrenalectomy.  When is it indicated?  Has any papers addressed this?  Surgery, 112:994-1001, 1992.

Author Response

Thank you very much for your sound and constructive comments and for giving us the opportunity to review and improve our manuscript.

In the attached file can find a detailed point by point answer to your comments and concerns.
